# The adaptive immune response to *Trichuris* in wild versus laboratory mice: An established model system in context

Iris Mair[1,2]☉*, Jonathan Fenn[3]☉*, Andrew Wolfenden[3], Ann E. Lowe[3], Alex Bennett[1], Andrew Muir[1], Jacob Thompson[1], Olive Dieumerci[1], Larisa Logunova[1], Susanne Shultz[4], Janette E. Bradley[3‡]*, Kathryn J. Else[1‡]*

1 Lydia Becker Institute of Immunology and Inflammation, School of Biological Sciences, Faculty of Biology, Medicine and Health, University of Manchester, Manchester, United Kingdom, 2 Manchester Environmental Research Institute, Department of Earth and Environmental Sciences, Faculty of Science and Engineering, University of Manchester, Manchester, United Kingdom, 3 School of Life Sciences, University of Nottingham, Nottingham, United Kingdom, 4 School of Natural Sciences, Faculty of Science and Engineering, University of Manchester, Manchester, United Kingdom

☉ These authors contributed equally to this work.
‡ JEB and KJE also contributed equally to this work.
* iris.mair@manchester.ac.uk (IM); Jonathan.Fenn2@nottingham.ac.uk (JF); jan.bradley1@nottingham.ac.uk (JEB); kathryn.else@manchester.ac.uk (KJE)

**Data Availability Statement:** All relevant data are within the manuscript and its Supporting information files.

## Abstract

Laboratory model organisms have provided a window into how the immune system functions. An increasing body of evidence, however, suggests that the immune responses of naive laboratory animals may differ substantially to those of their wild counterparts. Past exposure, environmental challenges and physiological condition may all impact on immune responsiveness. Chronic infections of soil-transmitted helminths, which we define as establishment of adult, fecund worms, impose significant health burdens on humans, livestock and wildlife, with limited treatment success. In laboratory mice, Th1 versus Th2 immune polarisation is the major determinant of helminth infection outcome. Here we compared antigen-specific immune responses to the soil-transmitted whipworm *Trichuris muris* between controlled laboratory and wild free-ranging populations of house mice (*Mus musculus domesticus*). Wild mice harbouring chronic, low-level infections produced lower levels of cytokines in response to *Trichuris* antigen than laboratory-housed C57BL/6 mice. Wild mouse effector/memory CD4+ T cell phenotype reflected the antigen-specific cytokine response across the Th1/Th2 spectrum. Increasing egg shedding was associated with body condition loss. However, local *Trichuris*-specific Th1/Th2 balance was positively associated with worm burden only in older wild mice. Thus, although the fundamental relationships between the CD4+ T helper cell response and resistance to *T. muris* infection are similar in both laboratory and wild *M. m. domesticus*, there are quantitative differences and age-specific effects that are analogous to human immune responses. These context-dependent immune responses demonstrate the fundamental importance of understanding the differences between model and natural systems for translating mechanistic models to 'real world' immune function.

**Funding:** This work was supported by the Biotechnology and Biological Sciences Research Council, grant number BB/P018157/1 awarded to KJE, and grant number BB/P017827/1 awarded to JEB (https://www.ukri.org/councils/bbsrc/). IM was supported by a Wellcome ISSF and University of Manchester EDI Perera Fellowship (grant number 204796/Z/16/Z) during part of the analysis and writing of the manuscript (https://wellcome.org/). The funders had no role in study design, data collection and analysis, decision to publish, or preparation of the manuscript.

**Competing interests:** The authors have declared that no competing interests exist.

## Author summary

Most knowledge of immune responses to parasitic infection stems from controlled laboratory mouse models. We found that adaptive immune responses to *Trichuris muris* under natural conditions are not fully reflected by laboratory infection models employing a single bolus of high or low egg doses. Whilst the Th1/Th2 paradigm persists in the wild, immune responses are dampened compared to laboratory mice. Further, we reveal that in wild mice, worm burden is only explained by the immune response in older, but not younger mice, a pattern previously observed in humans but not laboratory mice. By studying immune responses in a wild system, we can expand our well-established laboratory disease models to reflect the real world and enhance their utility for biomedical research.

## Introduction

The immune system is a highly complex and adaptable system, shaped throughout a lifetime by lifestyle and environmental factors. Unsurprisingly, the immune system of 'naive' adult laboratory mice resembles that of human newborns more than that of human adults [1]. It is therefore important to cross-validate laboratory models within ecologically relevant frameworks, and delineate associations in complex, natural systems that may not be possible to be recapitulated in the laboratory setting [2]. Indeed, there are now multiple reports pointing to wider environmental influences being critical in determining infection outcome and health impacts of experimental infections [3,4].

Measurement of functional adaptive immune responses in humans or wild animals is challenging. In humans, there are obvious ethical constraints limiting sampling to minimally-invasive studies, and human studies are logistically and financially particularly challenging. For wild non-model species, immunological analysis tools are often limited [5–7], and ethical considerations can limit study designs. Despite these difficulties, several studies have been performed recently to understand variation in T cell immune responses in wild mammals [8–13]. Further, pioneering studies have started to probe immune responses to parasite infection in *M. m. domesticus* in the wild [14] and laboratory mice released into outdoor enclosures [4]. However, no study in wild mammalian populations has quantified the local parasite-specific CD4+ T cell response which is deemed key to infection outcome, its association with infection in an uncontrolled setting, how the wild parasite-specific immune response compares to laboratory models currently used for biomedical research, and what the ultimate ecological consequences are for a given host animal.

We present a study that, for the first time, probes the local immune response to the Soil-Transmitted Helminth (STH) *Trichuris muris* in a naturally infected wild host and the potential health implications of individual variation in infection-immunity dynamics. STHs affect virtually all mammalian species, including humans, livestock and wildlife. Both health and economic concerns drive research into understanding disease susceptibility and potential ways to minimise disease burden both on the individual and population level. By every measurable health statistic, low/low-middle income countries are disproportionately affected by STHs, which include the gut-dwelling nematode parasite *Trichuris trichiura* [15]. Studies of peripheral immune responses to STHs in humans reveal a complex picture, with low dose chronic infections the norm and a potential build-up of Th2-mediated immunity after decades of infection exposure [16,17]. Infections are typically overdispersed within a population [18,19], an epidemiological pattern mirrored in wild rodents [20]. In order to better understand host-

pathogen interactions and to drive novel therapeutic development, experimental work in laboratory mouse models has been employed for several decades. A particularly useful helminth model species is *T. muris*, a natural parasite of *M. musculus*, which has high genome conservation with *T. trichiura* [21,22]. Laboratory studies employ largely single dose infections, often classified as either high dose (>100 eggs) or low dose (<40 eggs), in highly controlled environments. Such laboratory-based systems have demonstrated the importance of Th2 responses for the successful expulsion of gastrointestinal (GI) nematodes [16,23,24]. Subsequent studies in the laboratory mouse model have gone on to show that environmental context and host-associated factors are important in determining the quality of the anti-parasite immune response and thus infection outcome. For example, infection regime [25], diet [26,27], sex [28] and age [29] each play a significant role in shaping resistance to infection in the laboratory setting.

To allow in-depth immunological analysis, coupled with natural ecological variation, we studied a fully wild and isolated island population of house mice living with minimal human interference, using longitudinal and cross-sectional sampling. Given the multitude of host-intrinsic and -extrinsic factors at play in the wild, we hypothesised that factors shown to determine the immune response and susceptibility to *Trichuris* infection in the lab would have less of an impact in the wild. However, we further hypothesised that despite the 'noise' of this uncontrolled study system, we would be able find evidence of a link between T helper cell polarisation and disease susceptibility. In this study, we show that the local recall cytokine response is quantitatively significantly weaker in wild mice compared to those in a laboratory experimental setting, but showed similarities in the Th1/Th2 balance of their recall cytokine response. Of several parameters measured, only age and worm burden explain, to a small degree, local anti-parasite cytokine responses. Importantly, only in older, but not younger wild mice, did we find the evidence of a link between T helper cell polarisation and worm burden. Furthermore, longitudinal data analysis revealed a connection between changes in infection levels and host health. Overall, our study population mimicked several parasite-immune relationships found in human populations where *Trichuris* is endemic; and highlights that the uncontrolled environment likely supersedes several of the mechanisms found in laboratory settings. We suggest that by studying the immune response in a wild house mouse system alongside more controlled and mechanistic laboratory studies, we can expand our well-established laboratory disease models to include an ecologically relevant context and thereby enhance their utility for translational biomedical research.

## Results

### The Isle of May mice harbour chronic *Trichuris muris* infections, presenting a measurable health impact at high infection levels

Worm burden was assessed upon cull of 272 wild mice, revealing a *Trichuris* prevalence of 79%. ITS1-5.8S-ITS2 rDNA sequencing confirmed the parasite species to be *T. muris*, with 100% sequence overlap with the Edinburgh (or E) isolate used in laboratory studies for decades [30] (S1 Fig), and closely associating with other Western European isolates previously reported in wild-caught rodents [31] (S2 Fig). The majority of mice harboured low levels of *Trichuris* (<10 adult worms) (Fig 1A), in an overdispersed distribution pattern, with mean and median worm counts of infected individuals being 11.5 and 5, respectively. Animals without worms but seropositive for anti-*Trichuris* IgG1/IgG2a represented 20% of the study population. Only 1% of animals were both free of worms and seronegative. Although the majority of animals that were visually classified as juveniles already presented with worms, the prevalence of worm infections in adults was significantly higher than in juveniles (Fig 1B). Using a logistic regression model and a proxy for age, derived from a primary principal component capturing multiple aspects of morphometry expected to increase with biological age (hereafter 'maturity

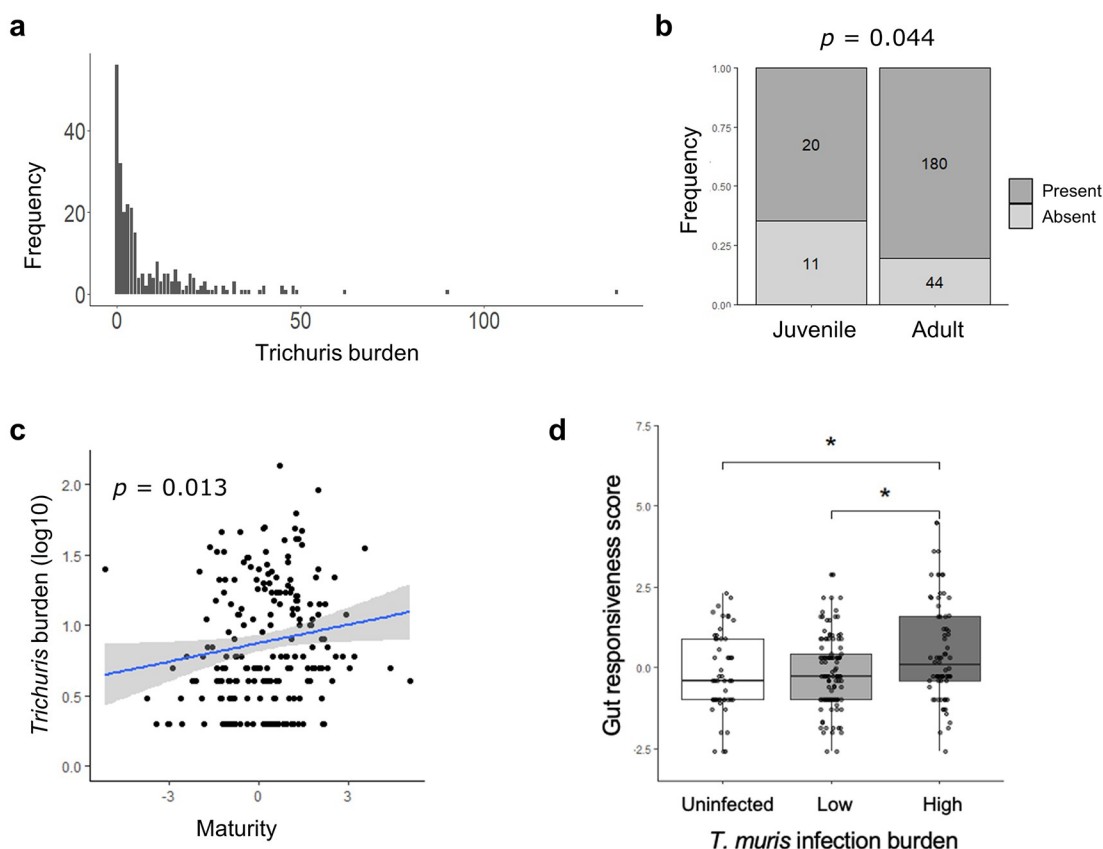

**Fig 1.** ***Trichuris muris*** **is a highly prevalent endoparasite in Isle of May mice causing mainly chronic infection. a)** Frequency distribution of *Trichuris* burden in wild caught *Mus musculus* on the Isle of May, assessed microscopically *post-mortem*. **b)** Prevalence of Trichuris infection in juveniles and adults, analysed using a Pearson's Chi-squared test. **c)** Correlation plot of Trichuris burden and maturity index as proxy for age, with p value derived from a generalised linear model explaining Trichuris burden with maturity index, sex, date of capture and scaled mass index (accounting for year of capture). **d)** Group-wise comparison of gut responsiveness score between uninfected individuals, low level infected individuals (1–9 worms) and higher level infected individuals (>9 worms) (Kruskal-Wallis test followed by a Dunn's test). *p≤0.5, **p≤0.01, ***p≤0.001.

index', see Methods for further details), we confirmed that older animals had a significant higher chance of being infected than younger mice (logistic regression odds ratio = 1.446, $p = 2.13 \times 10^{-4}$). Furthermore, maturity index was the only factor positively associated with worm burden in a generalised linear model (GLM) (Fig 1C, see S1 Table for full model output), i.e. animals tend to accrue rather than expel worms with increasing age. Overall, these data support the hypothesis that exposure to *Trichuris* infection is high in this wild mouse population, and infections are chronic in nature.

As is the case with many free-living animals and affected human populations, the Isle of May mouse population was polyparasitised. In addition to *T. muris*, two further helminth endoparasites were found as described in a previous study [32]. Specifically, the gut-dwelling pinworm *Syphacia obvelata* and the liver nematode *Calodium hepaticum* were found in a number of animals over the duration of this study with a low prevalence of *S. obvelata* and high prevalence of *C. hepaticum* (S3 Fig). *C. hepaticum* can be transmitted via necrophagy, or through ingestion of embryonated eggs in the environment following decay of an infected host [33]. There is no explicit evidence, nor anecdotal report of necrophagy occurring in this population (previous studies found variation in sources of dietary nitrogen, but the source of this

variation is unknown [34]), and as such this parasite has been assumed to be transmitted through environmental ingestion in this instance. Whilst immunomodulation by *Calodium* is likely, we did not carry out further parasite-specific assays due to the scarcity of diagnostic tools for this parasite [35]. To assess the health consequences of harbouring *T. muris* at the organ level, we generated a proxy for gut architecture disturbance comprised of the primary principal component of four histological measures (level of immune cell infiltration, location of infiltration, goblet cells per crypt, and crypt length) which we termed 'gut responsiveness' (see Methods and S4 Fig for further details). *Trichuris* burden was positively associated with increased gut responsiveness at worm burdens of 10 or higher (Kruskal-Wallis $p$ = 0.023), suggesting that infection level is an important determinant of the impact of infection on host health (Fig 1D).

## The cytokine recall response to *Trichuris* in wild mice is significantly weaker than in laboratory single dose infection models

We next quantified the local adaptive immune responses to parasite exposure of wild mice living in a natural environment versus those of laboratory mice. To this end, upon dissection, mesenteric lymph node cells of wild mice and experimentally infected laboratory mice underwent recall stimulation with parasite excretory-secretory product (E/S) and secreted cytokines were measured (Fig 2A–2C). A principal component analysis (PCA) was applied to wild mouse data to reduce the complexity of the multi-cytokine response (Fig 2D). Principal component 1 (PC1), capturing 42% of the variation in cytokine expression, denoted increasing cytokine concentrations overall. Whilst no discrete clusters were apparent within the wild mouse population, the loadings of PC2 (18% of variation) clearly pointed towards animals shifting along an axis of either a more Th2-dominated (IL-4, IL-5, IL-9, and IL-13) or a regulated Th1/pro-inflammatory response (TNF-a, IL-6, IFN-g, IL-17, IL-10). Strikingly, cytokine responses did not differ between wild mice harbouring *Trichuris* worms or not, at the time of analysis, in the context of both response strength (PC1) and response quality (PC2) (Fig 2E). Inclusion of laboratory animals in the analyses revealed that the cytokine responses of the majority of wild and laboratory animals overlapped at least partially (Fig 2F), with the notable exception of laboratory mice culled at day 34 post high dose infection, at a time point when worms have likely been expelled under this experimental regime. Using a Kruskal-Wallis test followed by a Dunn's test with Bonferroni adjustment to compare PC1 and PC2 across experimental groups, wild mice overall secreted significantly lower levels of cytokines upon recall than laboratory mice under any of the infection regimes and time points (Fig 2G). For reference, naive lab mice were included as controls in a separate PCA and exhibited a lower median for overall cytokine expression in response to *Trichuris* E/S (PC1 median: -1.3193, IQR: -2.4500, -0.3607) compared to wild mice (PC1 median: -0.2544, IQR: -1.3020, 0.8524). PC2, or the Th1/Th2 balance of the parasite-specific immune response, in wild animals overlapped fully with C57BL/6 mice at day 21 and day 34 following a low dose infection. In summary, whilst the Th1/Th2 balance of the anti-*Trichuris* cytokine response in wild mice was comparable to a laboratory chronic infection model, mLN cells of wild mice overall secreted less cytokine in response to recall stimulation with *Trichuris* E/S product.

## CD4[+] T effector/memory cell phenotype is linked to antigen-specific cytokine production in wild mice but does not explain differences between wild and laboratory immune responses

Since the main source of cytokines in the recall response are CD4[+] T cells, we next performed flow cytometric analysis to assess whether different T cell phenotypes explained the observed

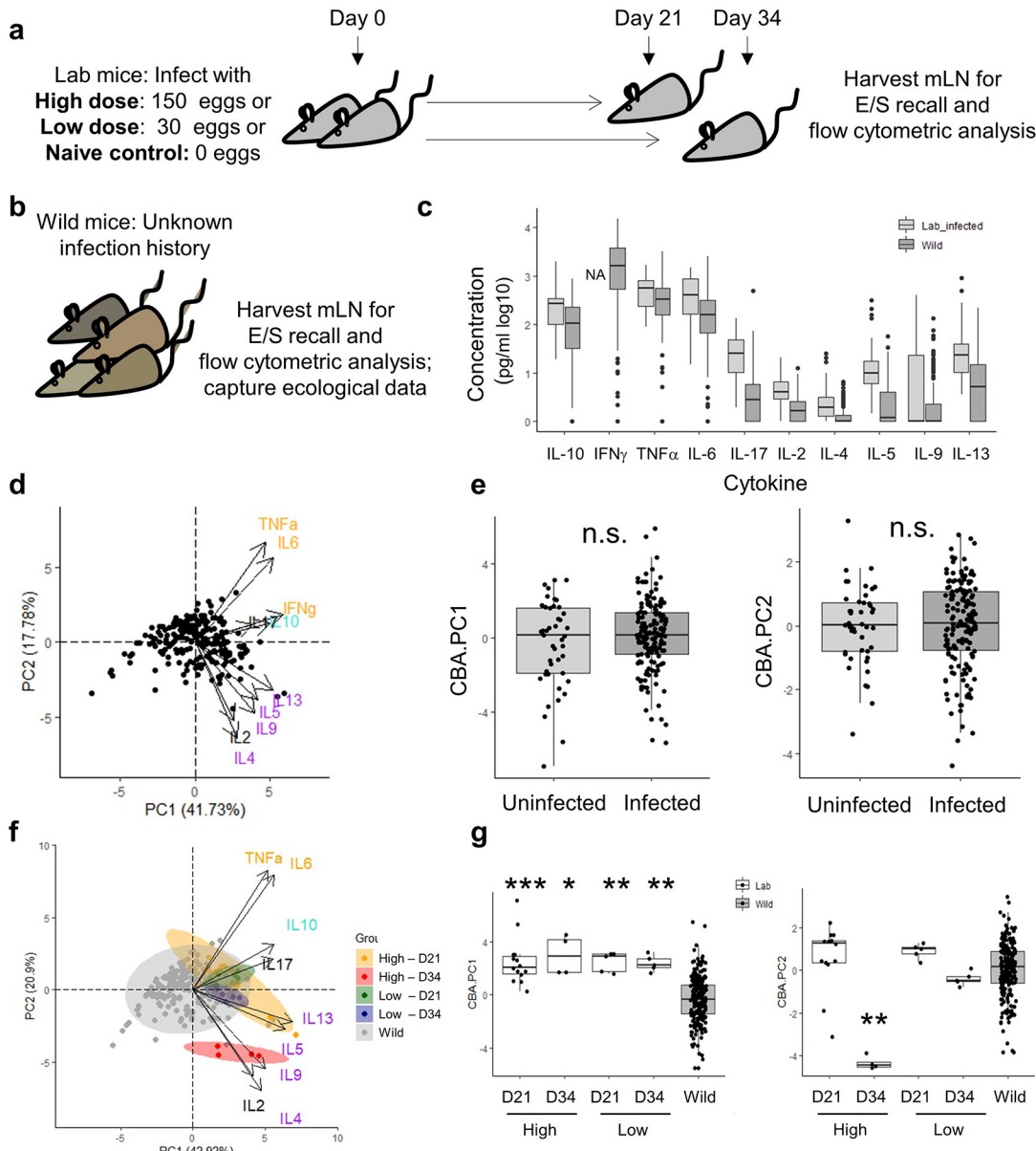

**Fig 2. Wild mice vary in their local anti-*Trichuris* Th1/Th2 balance and secrete lower levels of cytokines than laboratory mice. a)** Experimental design for *T. muris* infection in laboratory mice. **b)** Experimental design in wild mice. **c)** Cytokine concentrations in cell supernatants of mesenteric lymph node cells cultured for 48h in the presence of *T. muris* E/S for 48h. **d)** Principal components 1 (PC1) and 2 (PC2) of a principal component analysis of the parasite-specific cytokine response in wild mice. **e)** PC1 and PC2 of PCA analysis from (D) in uninfected versus infected wild mice. **f)** Principal components 1 (PC1) and 2 (PC2) of a principal component analysis of the parasite-specific cytokine response in wild mice and laboratory mice experimentally infected with *T. muris*. **g)** PC1 and PC2 of PCA analysis from (F) grouped by experimental group and analysed using a Kruskal-Wallis test followed by a Dunn's test with Bonferroni adjustment. Statistical significance only presented for comparison of wild versus laboratory groups. *p≤0.5, **p≤0.01, ***p≤0.001.

variation in cytokine expression in mLNs of wild animals (S5 Fig (gating), Fig 3A). Hypothesising that the proportion of CD4+ T cells, Foxp3$^+$ regulatory T cells, T$_{EM}$, T-bet$^+$ T$_{EM}$ and GATA3$^+$ T$_{EM}$ may affect the observed variation in cytokine expression, we used these markers as predictors in a generalised linear models to explain either PC1 or PC2. The proportion of T$_{EM}$ within the CD4$^+$ T cell pool was the only parameter positively correlated with PC1

(cytokine level) (Fig 3B), out of six parameters included in the GLM (see S2 Table for full model output). Thus, animals with a generally heightened T cell activation status in the mLN, as determined by flow cytometry, produced more cytokines in response to an antigen-specific challenge. PC2 (Th1/Th2 balance) was significantly associated with higher proportions of T-bet and lower proportions of GATA-3 expressing effector Th1 and Th2 cells respectively, following the expected Th1/Th2 paradigm [36].

To check whether differential cytokine secretion could be explained by underlying differences in the cellular make-up of the mesenteric lymph nodes, we also compared the relative percentages of CD4$^+$ T cell subsets in wild and laboratory mice using a Kruskal-Wallis test followed by a Dunn's test with Bonferroni adjustment. Lower cytokine expression levels in wild mice was not due to lower proportion of CD4$^+$ or T$_{EM}$ present in mLN (Fig 3Ci and 3Cii), or higher Treg proportions compared to laboratory mice (Fig 3Ciii). In fact, wild mouse mLNs harboured significantly enlarged populations of T$_{EM}$ compared to laboratory mice, as has been previously reported in other wild mouse populations and is considered likely a result of

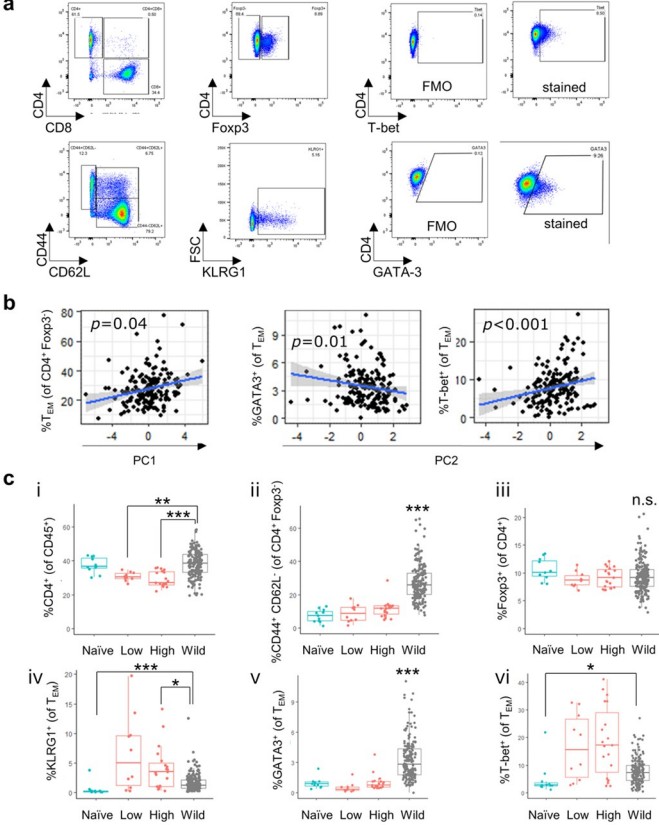

**Fig 3. CD4$^+$ effector T cell phenotype is linked to cytokine production but does not explain differences in anti-Trichuris cytokine response between laboratory and wild mice. a)** Representative flow cytometry plots showing gating strategies for CD4$^+$ T cells, Foxp3$^+$ regulatory T cells, effector memory CD4$^+$ T cells (T$_{EM}$), and T-bet and GATA3 expression in T$_{EM}$, in mesenteric lymph nodes (mLN) of wild mice. **b)** Correlation plots of PC1 and PC2 (from Fig 2D) with selected flow cytometric mLN CD4$^+$ T cell phenotypes; p values are derived from generalised linear models explaining either PC1 or PC2 with proportion of CD4+ T cells, Foxp3$^+$ regulatory T cells, T$_{EM}$, T-bet$^+$ T$_{EM}$ and GATA3$^+$ T$_{EM}$. **c)** Group-wise comparison of selected mLN CD4+ T cell phenotypes in wild (grey) and laboratory mice (blue: naive; red: infected with low or high dose of *T. muris*) using a Kruskal-Wallis test followed by a Dunn's test with Bonferroni adjustment. Statistical significance only presented for comparison of wild versus laboratory groups. *p≤0.5, **p≤0.01, ***p≤0.001.

repeated exposure to immunogenic antigens [1,13,37]. The co-inhibitory receptor KLRG1 was expressed in small proportions of $T_{EM}$ both in laboratory infected and wild mice (Fig 3Civ; thus we saw no indication of heightened T cell senescence or inhibition in the wild [38]. Interestingly, we observed a Th2 propensity in wild mice based on markedly higher proportions of GATA-3 expressing $T_{EM}$ compared to naïve or infected C57BL/6 mice (Fig 3Cv), although this was not reflected by enhanced Th2 cytokine production (see Fig 2E and 2F). In contrast, T-bet$^+$ Th1 cells were enriched in wild mice only in comparison to naive laboratory mice (Fig 3Cvi). In infected laboratory mice, proportions of Th1 cells were highly variable, a pattern which was associated with a temporal boost in Th1 cells at day 21 (median: 21.3; IQR: 17, 33.3) which subsided back to near naive levels by day 34 following infection (median: 3.62; IQR: 2.43, 4.44), as previously reported on the level of T helper IFN-γ expression [39,40]. Differing CD4$^+$ T cell phenotypes exhibited by laboratory and wild animals were therefore insufficient to explain the differences in anti-*Trichuris* cytokine responses between laboratory and wild mouse groups.

## *Trichuris* burden and age are linked with the parasite-specific immune response in wild mice

Given the multiple host-intrinsic and host-extrinsic variables at play within our wild mouse population we asked whether certain ecological variables could explain the variation in cytokine response. A redundancy analysis (RDA), correcting for year of capture, showed that the selected explanatory variables all together explained 4.3% of the variation in cytokine expression (global RDA significance p = 0.001, see S3 Table for full model output). Nevertheless, worm burden and maturity index were significant contributors to variation in cytokine expression (Fig 4A). Importantly, higher worm burdens were associated with strong, regulated

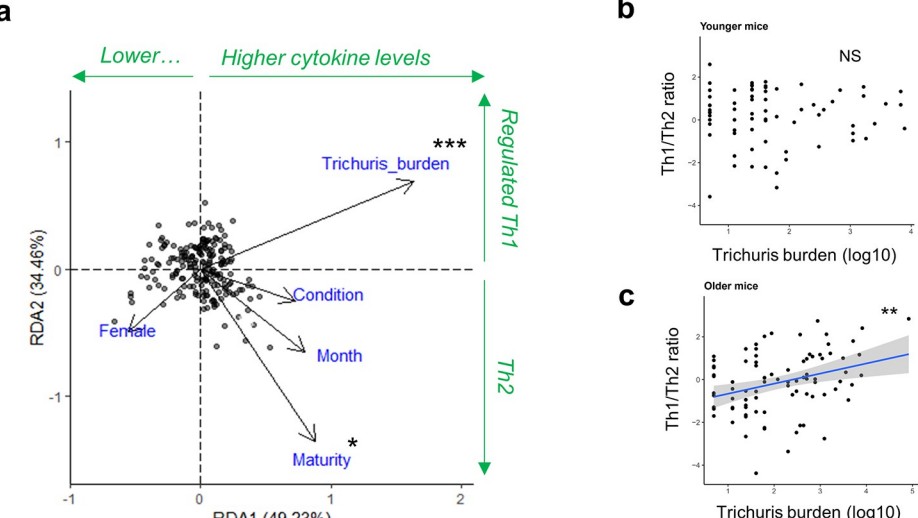

**Fig 4. Worm burden and age contribute to variation in anti-*Trichuris* immune response. a)** Redundancy analysis explaining variation in anti-Trichuris cytokine response in the mLN with ecological factors: Sex, maturity index as proxy for age, month of capture, scaled mass index as proxy for body condition, and *Trichuris* worm burden; with year and trap site accounted for. **b-c)** Scatterplot showing relationship between *T. muris* infection burden in infected individuals and relative strength of Th1 cytokine expression by mesenteric lymph node T cells measured by cytokine bead array, following stimulation with *T. muris* excretory-secretory molecules (using PC2 from Fig 2D). This data was separated into 2 equal cohorts of (b) lower and (c) higher maturity scores (age proxy). Explanatory terms provided to mixed model include *T. muris* burden (log counts+1), SMI, date as fixed factors, and sex as a random factor (NS = non-significant, *p≤0.5, **p≤0.01, ***p≤0.001).

Th1 responses (higher secretion of TNF-α, IL-6, IFN-γ, and IL-10). In contrast, increasing maturity explained the enhanced Th2 response (Fig 1D). The other variables including sex, body condition, and month of capture were not able to explain cytokine expression patterns.

The paradoxical finding that worm burdens generally increased with age (see Fig 1D), yet resistant Th2 responses also significantly increased with age warranted further investigation. Dividing the mice into equally-sized 'younger' and 'older' cohorts based on median maturity index (see S6 Fig for cutoff choice) showed that in younger mice, the Th1/Th2 balance did not distinguish between higher and lower levels of worm burden between individuals (Fig 4B). However, strikingly, in older individuals there was a positive association of worm burden with Th1 responses (mixed model $p = 0.0028$) (Fig 4B and 4C, see S4 Table for full model output). Thus, on the other hand, stronger Th2 responses were associated with lower worm burdens–in the older cohort only.

## Longitudinal data highlights temporal associations between host health, infection dynamics and immunity

A powerful tool in ecological studies is the use of longitudinal data to assess the relationship between host-intrinsic and -extrinsic factors over time. Using two key metrics for host health, body condition (SMI) and gut responsiveness, we aimed to assess potential health impacts of infection longitudinally. These two metrics were negatively associated (Spearman correlation, $p = 2.27 \times 10^{-6}$), and as such these analyses reflect different aspects of health in potential trade-offs. We leveraged longitudinal data of body condition, faecal egg counts, and serum anti-*Trichuris* IgG1 and IgG2a levels, of a subset of 91 mice, to probe associations between changes in these values and measures of animal health (see S8 Fig for raw data distribution). As with the worm counts, egg burden values showed an overdispersed distribution, though in individuals with both worms and eggs, there was little association between cull egg burden and worm burden (Spearman correlation, $p = 0.755$, n = 22). Body condition change was negatively correlated with egg burden change, in that animals with the strongest increase in body condition displayed the greatest reduction in egg burden over the course of a month, whilst animals losing body condition were more likely to display increased egg burden a month later (mixed model estimate = -0.73, $p = 0.006$) (Fig 5A). Gut responsiveness showed no association with changes in egg burden (mixed model $p = 0.29$), but did show a positive association with increases in anti-*T. muris* IgG1 (mixed model estimate = 0.37, $p = 0.008$). (Fig 5B, and for full model outputs see S5 Table).

## Discussion

Using a robust and extensive combination of immunological measurements and ecological data from over 200 animals, we have captured a comprehensive picture of the mammalian immune response to a natural infection with the parasitic nematode *Trichuris muris*. The prevalence of infection (>70%) in this island population of house mice is much higher than reported in many other wild rodent studies [41–43] but reflects that observed in some human populations where *T. trichiura* is endemic [17,44]. Th1/Th2-dependent immunity to *Trichuris* has been characterised in depth in laboratory mice [23,45,46], particularly using the C57BL/6 strain [25,27,29,39]. In order to better understand how well the laboratory based paradigms of resistance and susceptibility to helminth infection reflect immune responses in animals living in a multivariate environment, we compared our wild mouse data to that generated in C57BL/6 mice. Cross-sectional immunological analyses support general patterns of immunity observed from laboratory models, particularly a propensity for cytokine responses to vary along a Th1/Th2 axis, and a clear link between CD4$^+$ T$_{EM}$ cell phenotype and cytokine

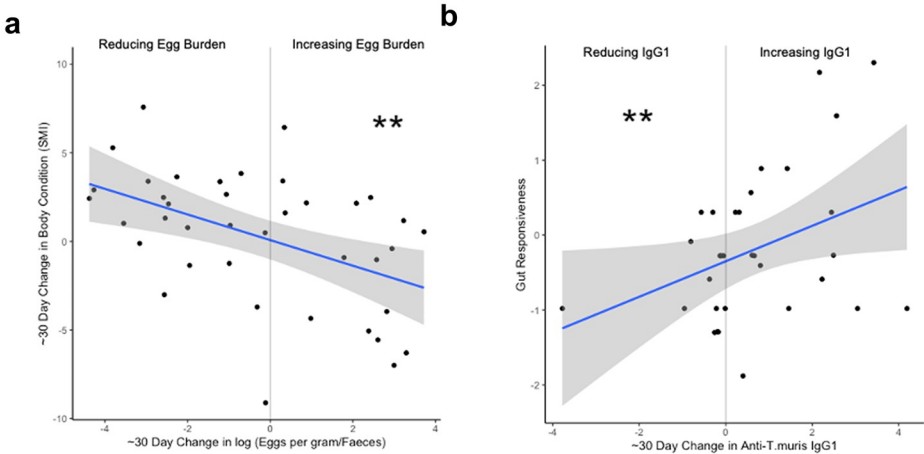

**Fig 5. Longitudinal shifts in infection status are linked to host health. a)** Scatter plot showing association between change in *T. muris* egg burden over 25–35 days and corresponding change in scaled mass index (SMI) as a measure of body condition (Mixed-effects model estimate = -0.73, p = 0.0054). **b)** Scatter plot showing association between change in anti-*T. muris* IgG1 over 25–35 days and cull gut responsiveness score (Mixed-effects model estimate = 0.37, p = 0.0085). Mixed-effects models considered only individuals with changing egg burdens, and predictors included change in egg burden, change in anti-*T. muris* IgG1 and IgG2a levels, with sex and age cohort as random effects (*p≤0.5, **p≤0.01, ***p≤0.001).

production. Novel insights include an age-associated dichotomy in the Th1/Th2 balance linked to difference in worm burden, as well as an overall reduced cytokine expression in response to parasite antigen compared to laboratory models.

The strength (cytokine levels) and quality (Th1/Th2 bias) were the major descriptors of variation in the *Trichuris*-specific immune response in wild mice, linked to the presence of effector T cells and their expression of T-bet/GATA3 in the mesenteric lymph nodes. Whilst the Th1/Th2 paradigm is well studied in laboratory parasite models, only few field studies have been able to confirm this immunological paradigm on a cellular level [9,12,14,16,17]. Cytokine responses in the context of *Trichuris* infection in mice–albeit not parasite-specific–have only been reported in two small-scale studies in natural [14] or semi-natural settings [47]. These studies point towards infected wild or re-wilded mice being more prone towards a Type 1 immune response, with heightened proportions of T-bet[+] cells and IFNγ production being reported. Indeed, the study by Zhang *et al.* [14] found no evidence of Th2 effector cells in mice infected with *Trichuris*, potentially–as the authors note themselves–due to the small size of the study cohort. In contrast, in the present study, we found evidence of a large pool of GATA3[+] effector T cells in the gut draining mesenteric lymph nodes, at levels higher than that seen in infected laboratory mice. Nevertheless, both cytokine responses and parasite epidemiological data from this study suggest that *T. muris* infection in the Isle of May mouse population is mainly of a chronic nature and sterile immunity is rare. Older mice overall harboured more worms than younger mice, as has also been reported in a wild *Apodemus sylvaticus* population [48]. In laboratory mice, ageing and immune senescence has been implicated in enhanced pro-inflammatory, Th1-dominated responses leading to higher susceptibility to infection [29]. Yet, we observe an overall increase in Th2 responses with age. It is noteworthy, however, that the majority of the mouse population on the Isle of May lives for less than one year due to population crashes each winter [49], and our study population therefore focuses on immune changes seen between juveniles and early adulthood, rather than senescence. Of the ecological variables tested to explain variation in cytokine response in wild mice, only age and worm

 

burden came out as significant, despite causative links of sex [28] and high-fat diet [27] with the immune response to this parasite in laboratory settings. These results showcase that experimental effects seen in laboratory conditions can be dampened or superseded by the abundance of other variation at play in the wild. Strikingly, in younger wild animals, Th1/Th2 balance was varied but did not associate with worm burden. However, in older wild animals, a link between Th2 responses and lower worm burden did emerge, as expected from laboratory studies. Studies in humans infected with soil transmitted helminths, including the human whipworm *T. trichiura*, have also described significant negative associations between chronic worm abundance and Th2 immune responses in an age-dependent manner [16,17]. Adopting the approach of dividing the study population into two age classes, both studies revealed that only in the older cohort, stronger Th2 immune responses were associated with reduced susceptibility, in keeping with our Isle of May study. Thus, it appears that in younger individuals in the wild–whether mouse or human–immune polarisation per se is not enough to control worm burden; only with increasing age, potentially through repeated infection exposure or changes in other host-associated factors, the Th1/Th2 balance becomes a determinant of susceptibility or resistance. Interestingly, in voles, it has been reported that older, but not younger animals display a tolerant phenotype of increasing worm burden concomitant with increasing body condition, linked with expression of GATA3 on a whole-blood RNA level [11]. This age-associated dichotomy has not been captured in laboratory mice thus far, where young animals are routinely used and display a clear link between Th1/Th2 balance and disease susceptibility. We would argue that in fact it may be hard to capture this dichotomy in comparatively small laboratory mouse studies due to the homogeneity of age, environmental variables, and exposure regimes. The Isle of May mouse population thus provides a tractable real world model for dissecting the underlying immune mechanisms at play in determining resistance and susceptibility to infection.

Motivated by a desire to understand where the wild mouse cellular and cytokine responses fell in the context of typical laboratory based analyses of the cellular immune responses to whipworm infection, we compared cytokine profiles seen in *T. muris* infected C57BL/6 mice to those of the wild mice. We chose the C57BL/6 mouse for a number of reasons: C57BL/6 mice are the most widely used inbred strain of mouse in laboratory studies in general; it was the strain used to create the *Mus musculus* reference genome; there is a wealth of data concerning how it responds to *T. muris* infection; it permits the exploration of immune response to both a high dose and a low dose infection levels of whipworm. In the current study, in addition to genetics, we held constant many of the extrinsic and intrinsic factors such as sex, age, diet, which varied in the wild mouse population and each of which has been shown singularly to affect the immune response in laboratory studies [27–29]. We acknowledge that this is a reductionist approach and creates an asymmetrical comparison, however the approach does allow a comparison to be made between the most commonly used laboratory mouse and a fully wild *Mus musculus* population.

Overall, we observed a reduced cytokine response in wild compared to laboratory mice in response to parasite-specific stimulation of mLN cells, despite an enlarged CD4$^+$ T$_{EM}$ pool compared to laboratory mice, typical for wild or microbially-exposed laboratory mice [1,13,37,50]. Significantly lower cytokine responses in wild mice have been observed in response to innate immune stimulation, but not consistently in response to stimulation of the adaptive immune response [13,50]. Likewise, exposure of laboratory mice to wild mouse microbiome [1,51,52] or environmental antigens [53] have pointed towards a heightening of the adaptive immune response compared to clean laboratory mice. Searching for the underpinning mechanism, we did not find evidence of enhanced T cell senescence, an immunological mechanism by which effector T cell responses to the infection could be curtailed [54].

Whilst laboratory models cannot capture the complexity of the real world and work with immature immune systems [1,52], our data still support the relevance of laboratory mouse systems in serving as models for animal populations living in uncontrolled settings. Thus, wild mice did not cluster entirely separately from all laboratory mice; overlap was observed with laboratory animals currently harbouring infection. The Th1/Th2 bias of the immune response in wild mice was comparable to that of single bolus low-dose infected laboratory mice, despite low level infections in the wild mice likely the consequence of the ingestion of low numbers of eggs over time [55]. Laboratory infection models have attempted to replicate infection scenarios analogous to those assumed to occur in natural conditions for STHs. *Trichuris* trickle infection experiments revealed that slow accrual of small numbers of worms over time enabled an eventual threshold of worm burden to be reached, at which point the mouse switched from a more Th1-dominated to a Th2 phenotype accompanied by worm loss, in a highly genotype-dependent manner [25,27]. The precise control offered over the time of infection and its subsequent progression is of course also a strength of the laboratory model. For example, we were unable to distinguish between parasite-specific cellular immune responses mounted by currently infected hosts versus currently uninfected (but seropositive) hosts. The variable infection history and concomitant ecological variation appeared to supersede the difference in immune response expected from a currently infected and a currently uninfected individual.

In addition to our cross sectional data we also present a longitudinal infection dataset to highlight dynamic shifts occurring between an animal's health and infection status. Whilst immunomodulatory parasite products are being explored for their potential therapeutic effect in autoimmune and allergic disease [56], it is clear that chronic STH infections can cause a significant health burden in affected individuals [15]. Time-series egg-count datasets have been used to elucidate fitness trade-offs associated with immunity to helminth infection and predict survival through crash years in wild mammals [57,58]. While longitudinal analyses can ideally make the directionality of these immune dynamics clearer, the results from analyses of changes between two time points showed little association between immunity and faecal egg counts, over time. The fact that egg counts are not a strong indicator of worm burdens limits the usefulness of these analyses if egg counts are used as a proxy for worm burden, and should be kept in mind for other field studies using egg burdens as an indicator of infection levels. While taking these limitations into account, alongside the fact that egg shedding can vary throughout the day [59], we suggest that repeated measures of faecal egg burden still provide a valuable insight into parasite fitness.

Bearing in mind the use of egg burden as a proxy for worm burden, the increase in body condition concomitant with a decrease in faecal egg burdens over time indicates that infection may confer some level of fitness cost in the mice, with measurable health benefits of reducing egg shedding [58,60,61]. The positive association observed between increasing levels of anti-*T. muris* IgG1 and gut responsiveness at cull suggests that driving of IgG1 levels, likely an indicator of stronger Th2 dominated immune responses [62], may come at the cost of local tissue damage. Th2 immune responses can induce both tissue regenerative and pathological processes [63]; especially in the wild, it is likely that trade-offs are at play between immunity and associated energetic costs and tissue damage [64]. Further longitudinal work, with multiple time points per animal would help to elucidate the directionality of relationships between immunity, body condition and infection.

To conclude, laboratory mice and wild mice differ in multiple ways each of which can influence their immune "phenotype". For example, laboratory mice do not have to contend with natural pressures such as temperature shifts, food scarcity, reproduction, or co-infections [65], and the immune system of laboratory animals is immature in phenotype and function compared to 'dirtier' animals [1,37,51–53]. Diversifying experimental parameters such as sex and

age, microbial exposure of laboratory mice, outdoor housing of laboratory mice, or the use of fully wild mouse populations, are therefore key additional approaches to investigating the immune system in order to develop a clear understanding of its complexity and function [2]. Questions arising from the current study include how much of the variation in immune response is genetically driven. Teasing apart genetic versus environmental drivers could be approached by genetic analysis of the wild mice and a comparison of immune responses across a variety of inbred and outbred laboratory strains in relation to the wild mice. Further, given that exposure to infection is likely repetitive in the wild, trickle infections in laboratory animals [25] would warrant a comparison. Finally, ecoimmunological studies such as the one we present here, serve as hypothesis generators for interventional experiments in the wild, targeting e.g. the diet or the immune system itself, in order to provide mechanistic insights. Studies in a wild sheep population have demonstrated that endoparasite worm burdens predict winter survival [58] and in human populations there is clear evidence of reduced quality of life in individuals infected with *Trichuris trichiura* [15]. What constitutes an appropriate immune response to parasites to minimise adverse effects on health–either due to exacerbated immune investment or due to costs of parasitism–and how these immune responses are shaped, are two of the big outstanding questions to be addressed in free-living or re-wilded populations.

## Materials and methods

### Ethics statement

The work on wild mice was approved by the University of Nottingham Animal Welfare and Ethical Review Body and complies with the UK's Animals (Scientific Procedures) Act of 1986. The work on laboratory mice was approved by the University of Manchester Local Animal Welfare and Ethical Review Body and complies with the UK's Animals (Scientific Procedures) Act of 1986.

### Mice

Wild house mice were live-trapped between August and December in 2018 and 2019 on the Isle of May (56°11′11·6″N, 2°33′24·1″W) on three independent trapping grids: 'Low Light' (LL), 'Main Light' (ML & 'Fluke Street' (FS). Each grid consisted of 96 Longworth traps, placed 8–10 metres apart in a 6 × 16 grid and containing Sizzle Nest (Dates and, catalogue number CS1A09) and sunflower seeds, as described in Mair *et al.* 2021 [66]. August to December was chosen as trapping season as the mice undergo a yearly population crash over winter with sufficient trapping power only returning around August each year. Trapping at one grid (ML) was halted following the August 2019 trapping session, due to limitations in field worker numbers. There were a total of 14 trapping days at the ML grid, 30 at the LL grid and 31 at the FS grid (trapping effort detail provided in S9 Fig). From August to December each year, caught mice were checked for Passive Integrated Transponders (PITs), and were PIT tagged if none was detected. Basic morphometric measures (body mass, snout-vent body length and tail length) were taken using callipers and scales. A total of 492 mice were PIT tagged between August-November in 2018 and 2019. Mice captured from November to December in 2018 and from September to December 2019, and which had been previously trapped, were culled (n = 214). Another 58 untagged mice caught during culling sessions were also culled, where fieldwork capacity allowed. All culled mice formed our 'Cross-Sectional' set (n = 272), and were used for detailed immunological and histological analyses. Of these culled mice, 91 animals had been captured 30 days previously (recaptures) and so also entered into our 'Longitudinal' study, allowing us to investigate changes in individuals over time. Across the two years, overall recapture rate was 40% within one month.

Male C57BL/6 mice were bought from Envigo and maintained at a temperature of 20–22˚C in a 12-h light–12-h dark lighting schedule, in sterile, individually ventilated cages in same-sex groups of 2–5, with food and water ad lib. Laboratory mice were 8–13 weeks old when used for this study. All animals used for this study were euthanized by a rising concentration of $CO_2$.

### *Trichuris* DNA extraction and PCR sequencing

Genomic DNA from 3 female worms belonging to the Isle of May and Edinburgh *T. muris* isolates (6 in total) were extracted via a DNeasy Blood and Tissue kit (Qiagen) as per the manufacturer's instructions. Quality of DNA extractions were determined via a 0.8% agarose gel stained with SYBR Safe (Thermofisher). The ITS1-5.8S-ITS2 rDNA regions of each worm was then amplified via PCR using the nematode specific NC5: GTAGGTGAACCTGCGGAAG-GATCATT and NC2: TTAGTTTCTTTTCCTCCGCT primer set. Each PCR included 0.5 µl of each primer (0.5 µM), 12.5 µl of OneTaq Quick-Load 2X Master Mix with Standard Buffer (New England Biolabs), 2 µl of diluted (1:10) genomic DNA, and 10 µl of MilliQ water. PCR cycling parameters consisted of initial denaturation at 94˚C for 5 mins, followed by 35 cycles of 94˚C for 30 seconds, 54˚C for 30 seconds, and 72˚C for 60 seconds, accompanied by a final elongation step of 72˚C for 10 minutes. Obtained PCR products where then isolated on a 1% agarose gel stained with SYBR Safe (Thermofisher). DNA bands pertaining to the amplified PCR products were cut out of the gel under UV light and extracted from the agarose via a Qia-quick gel extraction kit (Qiagen) according to the manufacturer's protocols. Concentrations of the extracted PCR products were quantified via a Qubit 4 Fluorometer. Subsequently, obtained PCR products were diluted to 40 ng/µl in 10 µl of MilliQ water and underwent Sanger sequencing performed by the Genomic Technologies Core Facility at the University of Manchester.

### *Trichuris* phylogenetic analysis

All phylogenetic analysis was performed in MEGA11. ITS1-5.8S-ITS2 rDNA sequences of nematode species were aligned, and outlying regions were removed to make sequences uniform in length (977 bp) except for the distantly related species *Ascaris lumbricoides* and *Trichuris mastomysi* which were used to root the phylogenetic trees. The most appropriate phylogenetic model for each alignment of sequences was determined via the "*Find Best DNA/Protein Models (ML)*" function provided in MEGA11. For the sequence alignment underpinning S1 Fig, which contained ITS1-5.8S-ITS2 rDNA sequences pertaining to the *T. muris* Isle of May isolate alongside numerous other *Trichuris* species the maximum Likelihood method in tandem with the Tamura-Nei model and a discrete Gamma distribution to model evolutionary rate differences among sites (T92+G) was recommended. In contrast, only the maximum Likelihood method in conjunction with the Tamura-Nei model (T92) was suggested for the alignment associated with S2 Fig that consisted of many different *T. muris* haplotypes dispersed across Europe, published in Callejón *et al.* 2010 [31] and expanded by Wasimuddin *et al.* 2016 [67].

### Infection of laboratory mice with *Trichuris muris*

Mice were infected with *T. muris* eggs via oral gavage in a final volume of 200 µl of deionized water. For low-dose *T. muris* infection, 30 embryonated eggs were given to each mouse, and for high-dose infection, 100 infective embryonated eggs were given. Parasite maintenance, assessing of egg infectivity and counting of eggs were performed as described previously [68].

## Tissue preparation and cell isolation

Mesenteric lymph nodes were dissected and manually dissociated through 70 μm filters. Cells were counted using haemocytometers and 0.4% nigrosin (Sigma-Aldrich) dilutions for the exclusion of dead cells on the Isle of May, and using a CASY cell counter (Scharfe System) at the University of Manchester. Cells were re-suspended at $1x10^7$ cells/ml in RPMI 1640 buffer supplemented with 10% foetal calf serum (FCS, Sigma-Aldrich) for further analysis. The gastrointestinal (GI) tract was manually dissected, and a 0.3–0.4 cm snip of proximal colon fixed overnight in 10% neutral buffered formalin (Fisher), followed by long-term storage in 70% ethanol. Another 0.3–0.4 cm snip was fixed overnight in methacarn (64% methanol, 27% chloroform, 9% acetic acid), then transferred into 100% methanol for 30 minutes followed by 100% ethanol until processing. The residual gastrointestinal tissue was stored in 80% ethanol for later parasitological analysis.

## Mesenteric lymph node re-stimulation and cytokine bead array

Mesenteric lymph node cells were cultured in situ (on the Isle of May for wild mice, in the laboratory for laboratory mice) in flat-bottomed 96 well plates at $5x10^6$ cells/ml for 48h at 37°C 5% $CO_2$ in RPMI 1640 supplemented with 10% FCS, 1% L-Glutamine, and 1% penicillin/streptomycin (Invitrogen), with 50μg/ml of excretory-secretory product (prepared as detailed in [68]). Supernatants were harvested and stored at −20°C until cytokine determination. Supernatants from wild mice were stored at -80 degrees on the island prior to transport on dry ice to the University laboratories, where cytokine analysis was performed using the same assay and acquisition setup as for laboratory mice. Cytokines were analysed using the Cytometric Bead Array (CBA) Mouse/Rat soluble protein flex set system (BD Bioscience), which was used according to the manufacturer's instructions. Bead acquisition was performed on a MACSQuant (Miltenyi Biotech). For cytokine concentration analysis, FCAP Array v3.0.1 software (BD Cytometric Bead Array) was used. 10 cytokines were quantified in this assay: IFNγ, TNFα, IL-2, IL-4, IL-5, IL-6, IL-9, IL-10, IL-13 & IL-17.

## Flow cytometry

Single-cell suspensions from mesenteric lymph nodes were washed thoroughly with PBS and stained with Fixable Viability Dye eFluor 455UV and anti-CD16/CD32 (both from Thermo Fisher) in PBS for 10 min prior to addition of the relevant surface marker fluorochrome-conjugated antibodies in FACS buffer (PBS, 2% heat-inactivated FCS) supplemented with Super Bright staining buffer (Thermo Fisher). Following 30 min of incubation, cells were washed with FACS buffer and fixed Fix-Perm buffer of the eBioscience Foxp3 / Transcription Factor Staining Buffer Set (Thermo Fisher) overnight. Cell suspensions were kept at 4°C and in the dark throughout incubation steps. The following morning, samples were washed in Perm buffer of the same Buffer Set and stained with relevant intracellular marker fluorochrome-conjugated antibodies in Perm buffer for 30min at room temperature. After a final wash with Perm buffer, samples were re-suspended in FACS buffer and kept at 4°C in the dark until acquisition within 6 days of staining. The following antibodies were used: Ki-67-ef450 (clone SOLA15), Foxp3-ef506 (clone FJK-16S), CD62L-SB600 (clone MEL-14), CD44-SB645 (clone IM7), CD19-SB702 (clone EBIO1D3), CD4-SB780 (clone RM4-5), ICOS-FITC (clone C398.7), CD8-PerCP-ef710 (clone 53–6.7), ST2-PE (clone RMST2-33), KLRG1-PE-ef610 (clone 2F1), GATA-3-PE-Cy7 (clone TWAJ), T-bet-ef660 (clone 4B10), CD45-Af700 (clone 104), CD11b-APC-ef780 (clone M1/70). Samples were acquired on an LSRFortessa running FACSDiva 8 software (Becton Dickinson, Wokingham, UK). Data were analysed using FlowJo software (TreeStar; version 10.4.2). For gating strategy see S5 Fig.

## Serum antibody detection

Blood samples were taken from the tail vein for longitudinal sampling and/or after euthanasia via cardiac puncture. Serum was retrieved by centrifugation of clotted blood samples and stored at -80˚C prior to processing. Parasite-specific serum levels of IgG1 and IgG2c antibody were measured via ELISA. 96 well plates were coated with 5μg/ml *T. muris* overnight excretory-secretory product in PBS overnight, plates were washed, and non-specific binding blocked with 3% BSA (Sigma-Aldrich) in PBS. Plates were washed and incubated with serum dilutions (for IgG1 1:10,000; for IgG2 1:5,000) in triplicates. Each well also had a standard curve in duplicates using a single serially-diluted and aliquoted serum sample from a pool of *T. muris* infected laboratory mice across the entire study, allowing for inter-plate variability to be controlled for. Parasite specific antibody was measured using biotinylated IgG1 (BD Biosciences) or IgG2c (BD Biosciences) antibodies which were detected with SA-POD (Roche). After a final wash, plates were developed with TMB substrate kit (BD Biosciences, Oxford, UK) according to the manufacturer's instructions. The reaction was stopped using 0.18 M $H_2SO_4$ after 10 min incubation in the dark. The plates were read by a Versa max microplate reader (Molecular Devices) through SoftMax Pro 6.4.2 software at 450 nm, with reference of 570 nm subtracted. For each standard concentration on each plate, the mean OD value was calculated and a 4-parameter logistic model was fit across the serial dilution to form a standard curve for each plate. These standard curves were then used to estimate concentrations from OD values for the mouse blood serum wells. From sample triplicates, the greatest outlier was removed and the average of the remaining two values calculated. These averaged concentration values were then used for any downstream analysis.

## Parasitology

Prior to dissection of the ethanol-fixed gastrointestinal (GI) tract, the liquid in the tube was examined in a scored petri dish under dissecting microscope, to count any worms which had moved out of the GI tract during storage and transportation. The GI tract was then separated into stomach, small intestine, caecum and colon. Each section was opened in a scored petri dish using dissection tools, and gently washed out with water to dislodge any material inside. The washed-out contents, and the tissue lining, were examined under dissection microscope, and any worms counted. Historically in laboratory based studies and within the current manuscript, the word "chronic" has been defined as an infection maintained in the host until the egg laying mature adult worms are present [69,70] i.e. the life cycle would be maintained in a natural population. Thus, the animal is susceptible to infection. Any host that expels the parasite prior to the development of the adults stage (32 days post infection) is deemed resistance and unable to harbour a chronic infection.

Faecal samples were collected during live trapping and at cull, and transferred directly to 1.6ml tubes containing sodium acetate-acetic acid-formalin (SAF) solution. Egg counts were made using a modified version of the McMaster flotation method, as follows; faecal samples were homogenised by manual disruption with a bent metal seeker, and then centrifuged at 400G for 10 minutes. The proportion by volume of the resulting pellet to the liquid SAF component was recorded. For each sample, a glass slide was divided into 10 equal sections. The pellet was again manually disrupted and homogenised, and in turn, 10μl of homogenised sample was pipetted across each slide section and checked under a microscope at x10 magnification. A cover slip was placed over the slide, and the number of *T. muris* eggs per section were counted and averaged. The average egg count per section and SAF-to-pellet volume ratio were used to calculate number of eggs per ml. All parasitological surveys were conducted at the University of Nottingham.

## Gut responsiveness score

A 'gut responsiveness score' was derived based on variation in colonic crypt length, extent of inflammatory infiltrate, depth of inflammatory infiltrate, and goblet cell numbers per crypt in wild mice. We chose an alternative phrase to 'gut pathology score', standardly used in laboratory colitis models or human IBD samples [71,72], as we cannot ascertain whether a higher score is a pathological response in wild animals. Scoring systems have previously been deployed by ourselves in the context of whipworm infection [73] and experimental colitis [74].

Neutral-buffered-formalin fixed tissues were dehydrated through a graded series of ethanol, cleared in xylol and infiltrated with paraffin in a dehydration automat (Leica ASP300 S) using a standard protocol. Specimens were embedded in paraffin (Histocentre2, Shandon), sectioned on a microtome (5μm sections) and allowed to dry for a minimum of 4 hours at 40˚C. Prior to standard H&E staining, slides were deparaffinised with citroclear (TCS biosciences) and rehydrated through alcohol (100% to 70%) to PBS or water. H&E stained sections were analysed using CaseViewer; crypt length was measured, and extent (graded 0–4) and depth (mucosal, submucosal, transmural) of inflammatory infiltrate assessed.

In order to measure goblet cells per crypt, methacarn-fixed tissues were cleared in xylol and infiltrated with paraffin in a dehydration automat (Leica ASP300 S) using a standard protocol prior to paraffin-embedding and tissue sectioning as describe above. Mucins in goblet cells were stained with 1% alcian blue (Sigma-Aldrich) in 3% acetic acid (Sigma-Aldrich, pH 2.5) for 5 mins. Sections were washed and treated with 1% periodic acid, 5 minutes (Sigma-Aldrich). Following washing, sections were treated with Schiff's reagent (Vicker's Laboratories) for 10 minutes and counterstained in Mayer's haematoxylin (Sigma-Aldrich). Slides were then dehydrated and mounted in DPX mounting medium (Thermo Fisher). For enumeration of goblet cell staining, the average number of cells from 20 crypts was taken from three different sections per mouse. Images were acquired on a 3D-Histech Pannoramic-250 microscope slide-scanner using a *20x/ 0.30 Plan Achromat* objective (Zeiss).

Four measures of gut responsiveness (crypt length, extent of inflammatory infiltrate, depth of inflammatory infiltrate, and goblet cell numbers per crypt) were used in a PCA to examine their covariance. PC1, explaining 43.4% variation, described a positive association between all 4 variables (S4 Fig). The PC1 eigenvalues for each individual were extracted for use as a multiparameter 'gut responsiveness score' in downstream analyses. A PCA was used in place of an additive multiparameter score in order to give a scaled, and less dimensional measure of gut responsiveness i.e. to remove the influence of any negative relationships between parameters, and leave only a measure of increasing responsiveness across parameters. These PC1 scores are highly correlated with the additive multiparameter score (Spearman's rho = 0.87, $p < 2.2e^{-16}$).

## Assessment of ecological variables and calculation of proxies

Sex of the animal was assessed visually at cull. Scaled Mass Index [75] was calculated from body length and mass as a measure of body condition relative to the whole population. Eyes were removed at cull and stored at room temperature in 80% formalin. Lenses were later dissected from eyes and dried in an incubator to remove any retained water for 5 days, the point at which the mass of eye lenses stopped decreasing. While age is typically estimated from dry eye lens mass in wild rodent populations, building on the formulae set out by Rowe *et al.* 1985 [76], we found this method to not be fully accurate on its own, as ages were calculated which were known to be impossible i.e. mice were known to be older than the calculated ages based on recapture data. In order to develop a useful and continuous proxy measure of age, a principal component analysis (PCA) was performed on cull samples, incorporating body mass, body length, tail length and dry paired eye lens mass, in order to provide a 'maturity index' (S6 Fig).

PC1 explained 69.6% of variation in these samples and described uniform increases across all included metrics, and so PC1 scores were taken as maturity index values. In order to have an equivalent index for non-cull samples, another PCA was carried out across all samples, excluding eye lens mass. PC1 explained 69.3% of variation in this analysis. These maturity PCA scores were strongly correlated with the analogous scores from cull (Pearson correlation estimate 0.976, $p<2.2e^{-16}$), meaning the longitudinal scores are a suitable proxy for the cull values which include eye lens mass data. Increases in longitudinal maturity index showed strong positive correlation with the time since an animal was first captured (Spearman correlation $p = 7.194e^{-15}$). The longitudinal maturity index was therefore taken as an appropriate measure of biological age when required for longitudinal analysis.

## Statistical analyses

General: All data were analysed using R version 4.1.2 (2021-11-01) [77]. Relevant statistical tests are described in results sections and figure legends(full model outputs: S2–S5 Tables).

Redundancy analysis: To explain how patterns of variation in cytokine profiles are explained by infection with *T. muris*, against a variable ecological background, while capturing any-multi-dimensional structure within the cytokine data, redundancy analysis was performed on cytokine concentration data from the cytokine bead array, using the R package '**vegan**' [78]. Cytokine concentration values were used as response variables, and *T. muris* infection burdens (log (count+1) values) SMI, trapping month, age (as maturity index estimate), and sex were included as explanatory variables, with year and trap site accounted for (n = 199). The redundancy model was tested for overall significance, as well as each axis of variation, and each explanatory variable.

Longitudinal analysis: For longitudinal analysis, linear mixed models were used to assess the influence of changing infection status and antibody levels on gut responsiveness and changes in body condition while controlling for inter-individual variation. A subset of the dataset was used for which two time points of faecal egg counts were available, and which had associated anti-*T. muris* antibody concentration data for both time points (n = 87). For all analyses involving egg counts, log (count+1) values were used. Values of change in SMI and anti-*T. muris* IgG1 and IgG2a were calculated (henceforth ΔSMI, ΔIgG1, and ΔIgG2a, respectively). Two linear mixed models were performed with change in body condition, and gut responsiveness at cull included as response variables, and individual animal ID included as a random factor. As we specifically aimed to model change in egg burden, we excluded any individuals with no eggs present in faecal samples at both time-points (remaining n = 40). Included as independent variables were change in log(eggs per gram + 1), ΔIgG1 and ΔIgG2a, with sex and age cohort (where maturity index <0 = "Young" and maturity index >0 = "Mature") as random factors. Age cohorts were chosen as we expected age-related changes in *Trichuris* immunity to be present as a sharp, qualitative shift, rather than a gradual linear relationship, with the cohort cutoffs selected to provide roughly equal group sizes for optimal model fitting, in lieu of a reliable measure of true age with which to establish a biologically-informed cutoff value (age-cutoff comparison and justification shown in S7 Fig). There were 124 mice designated to the 'young' cohort and 142 to the 'old' cohort. Group sizes were not identical as the median Maturity Index value of 0 was established from the whole cross-sectional dataset.

## Supporting information

**S1 Fig. Evolutionary history of whipworms isolated from house mice on the Isle of May.** The phylogenetic tree generated by the MEGA11 software is based on the maximum

Likelihood method in tandem with the Tamura-Nei model and a discrete Gamma distribution to model evolutionary rate differences among sites (T92+G) as selected by MEGA11 as the best fit model. The numbers below the branches represent nodes with >50% bootstrap support from trees generated from 1,000 bootstrap replicates and the maximum likelihood tree topology is scaled to the expected number of nucleotide substitutions per site and is defined by a scale bar (bottom left). The distantly related parasitic roundworm *Ascaris Lumbricoides* is used to root the tree. Sequences for known *Trichuris spp.* are labelled according to their species name followed by their GenBank accession number (e.g. AM234616.1). The sequence of the Edinburgh *T. muris* isolate is labelled by location and GenBank accession number (e.g. Edinburgh LC171641.1). The sequence for the unknown *Trichuris* species sampled from house mice on the isle of May is outlined in bold.
(TIF)

**S2 Fig. Phylogenetic analysis of the relationship between *T. muris* parasites found in house mice on the Isle of May and those sampled from house mice, rats, and wood mice throughout Continental Europe and island of Mallorca via comparison of ITS1, 5.8s, ITS2 regions in the ribosomal DNA of *T. muris* (Callejon *et al.* 2010) (31).** The phylogenetic tree was generated using the MEGA11 software. The evolutionary history between the parasites was inferred using the Maximum Likelihood method in conjunction with the Tamura-Nei model (T92) as selected by MEGA11 as the best fit model. The numbers below the branches represent nodes with >50% bootstrap support from trees generated from 1,000 bootstrap replicates. *Trichuris mastomysi* was used to root the tree. The maximum likelihood tree topology is scaled to the expected number of nucleotide substitutions per site and is defined by a scale bar (bottom left). The two Continental European clusters and Pan-European cluster identified by Callejon et al. [31] and Wasimuddin et al. [67], respectfully, are outlined to the right of the tree. The sequence from the Isle of May/Edinburgh strain of the parasite is highlighted in bold. Sequences derived from Callejon et al. [31] are labelled according to haplotype (e.g. H51), host species (As: Apodemus sylvaticus, Af: Apodemus flavicollis, Mmd: Mus musculus domesticus, Rr: Rattus rattus), and location (e.g. Turkey).
(TIF)

**S3 Fig. Co-infection rates for the three nematode parasite infections present in Isle of May Mice.** These include whipworm (*Trichuris muris*), pinworm (*Syphacia obvelata*) and the capillariasis-causing hepatic nematode *Calodium hepaticum*. Prevalence was confirmed through gastrointestinal dissection surveys of mature worms for *T. muris* & *S. obvelata*, and characteristic lesions and discolouration of liver tissue for *C. hepaticum*. Percentages indicate the proportion of the host population displaying a given combination of infections, with larger proportions of the population indicated by darker red colouration of the venn diagram.
(TIF)

**S4 Fig. Composite score for gut responsiveness.** Ordination plot showing principal component analysis (PCA) of gut responsiveness measures. PC1 describes an increasing level of gut responsiveness across all 4 included measures, and as such PC1 scores were extracted for use as a multiparameter 'gut responsiveness score' in further analyses. Methods of assessment of responsiveness components are included in methods and materials.
(TIF)

**S5 Fig. Flow cytometric gating strategy for CD4+ T cell subsets in wild mice.** Mesenteric lymph nodes were collected from wild house mice from the Isle of May between November 2018 and December 2019, and single cell suspensions stained for flow cytometric analysis.

Representative flow cytometry plots showing gating strategy for CD4$^+$ T cells (red box), Foxp3$^+$ regulatory T cells (Treg, orange box), effector memory CD4$^+$ T cells (T$_{EM}$) (purple box).
(TIF)

**S6 Fig. Maturity Indices for Age Approximation.** Ordination plot showing principal component analysis (PCA) of age-associated morphological traits taken from **a)** a cross-sectional cull dataset incorporating dry paired eye-lens mass, snout-vent body length, tail length and body mass and **b)** a longitudinal dataset from live-trapping, which omits eye lens mass. In both PCAs, PC1 explained the majority of variation, and described an increase across all measures. As such, PC1 scores were extracted for use as cross-sectional and longitudinal 'maturity indices' in further analyses.
(TIF)

**S7 Fig. Comparisons of mixed models testing effect of *T. muris* burden on MLN Th1/2 cytokine polarisation, with cutoffs for young and old age cohorts at different values of maturity index.** Cutoff values shown include the 'young' cohort comprising the bottom 10%, 30%, 46.62% (at maturity index = 0), 50%, 70% & 90%, with the remainder comprising the 'old' cohort in each case. **a)** Log p values of association between *T. muris* burden and MLN cytokine expression in the 'old' cohort models at different age cohort cutoffs, reaching a minimum at 46.62%. **b)** The residual maximum likelihood (REML) of 'young' and 'old' cohort models at different age cohort cutoffs, showing intermediate REML with least discrepancy between cohorts at 46.62%. (Note: 'Old' cohort models at 70% and 90% cutoffs gave overfitting errors when run).
(TIF)

**S8 Fig. Longitudinal data distribution summary.** Paired boxplots show changes between point of initial capture to 30 days later (± 5 days) within lines connecting individual mice, across four longitudinal measures: **a)** Serum anti-*T. muris* IgG1 concentration (Box-Cox normalised), **b)** serum anti-*T. muris* IgG2a concentration (Box-Cox normalised), **c)** scaled mass index, a measure of body condition and **d)** *T. muris* eggs per gram of faeces (log10 transformed, excluding samples with zero eggs). **e)** Distribution of faecal *T. muris* egg counts (excluding samples with zero eggs). **f)** Proportion of faecal samples where *T. muris* eggs are present versus absent.
(TIF)

**S9 Fig. Trapping schedule across the trapping grids.** Trapping grids named 'Fluke Street' (FS), 'Low Light' (LL) and 'Main Light' (ML) were used for fieldwork from August to December of 2018 (top) and 2019 (bottom). Each grid consisted of 6 rows of 16 Longworth traps. No trapping occurred at ML from September to December 2019.
(TIF)

**S1 Table. Summary statistics of *T. muris* worm burden GLM.** Each row corresponds to an explanatory variable provided to a generalised linear model, using log(*T. muris* worm burden +1) as the response (*p≤0.5, **p≤0.01, ***p≤0.001, n = 256).
(TIF)

**S2 Table. Summary statistics of flow cytometry GLM.** Each row corresponds to an explanatory variable provided to a generalised linear model, using **a)** mesenteric lymph node cytokine concentration PC1 (representing overall total concentration/strength of cytokine response) and **b)** PC2 (relative dominance of Th1 over Th2 cytokine concentration) as response

variables. (*p≤0.5, **p≤0.01, ***p≤0.001, n = 162).
(TIF)

**S3 Table. Summary of cytokine redundancy analysis. a)** Top—Summary statistics of each redundancy axis, Bottom–loadings of each axis, showing the relative contribution of concentrations of different cytokines to each axis. **b)** Summary statistics of explanatory variables provided for redundancy analysis. *p≤0.5, **p≤0.01, ***p≤0.001. SMI; Scaled mass index.
(TIF)

**S4 Table. Summary of age-separated cytokine-*Trichuris* burden mixed models.** Summary statistics for two mixed-effect models incorporating MLN cytokine PC2 (representing relative Th1 versus Th2 dominance) as the response variable, in juvenile (**left**: maturity index < 0) and mature (**right**: maturity index > 0) mice. Log(worm burden + 1), body condition (SMI) and date were included as fixed factors, and sex was included as a random factor. *p≤0.5, **p≤0.01, ***p≤0.001, n = 33. SMI; Scaled mass index.
(TIF)

**S5 Table. Summary of longitudinal mixed models.** Summary statistics for two mixed-effect models incorporating longitudinal data, with the response variables as ~30 day change in body condition (left) and gut responsiveness score (right). Age cohort (where maturity index > 0 equates to the 'mature' cohort, and < 0 the 'young' cohort) and sex were included as random factors. *p≤0.5, **p≤0.01, ***p≤0.001, n = 33. SMI; Scaled mass index.
(TIF)

## Acknowledgments

We would like to thank Nature Scotland for permission to carry out work on the Isle of May; David Steel (Nature Scotland), Bex Outram (Nature Scotland) and Mark Newell (Centre for Ecology and Hydrology) for support with fieldwork; and our many fieldwork volunteers. We thank Prof. John Brookfield (University of Nottingham) and Prof. Andrew MacColl (University of Nottingham) for their feedback on the manuscript drafts. We thank the University of Manchester Flow Cytometry Core Facility and the Genomic Technologies Core Facility for their technical support.

## Author Contributions

**Conceptualization:** Janette E. Bradley, Kathryn J. Else.

**Data curation:** Iris Mair, Jonathan Fenn, Andrew Wolfenden, Ann E. Lowe, Alex Bennett, Andrew Muir, Jacob Thompson, Larisa Logunova, Janette E. Bradley, Kathryn J. Else.

**Formal analysis:** Iris Mair, Jonathan Fenn.

**Funding acquisition:** Janette E. Bradley, Kathryn J. Else.

**Investigation:** Iris Mair, Jonathan Fenn, Andrew Wolfenden, Ann E. Lowe, Alex Bennett, Jacob Thompson, Olive Dieumerci, Larisa Logunova, Janette E. Bradley, Kathryn J. Else.

**Methodology:** Iris Mair, Jonathan Fenn, Andrew Wolfenden, Ann E. Lowe, Alex Bennett, Andrew Muir, Jacob Thompson, Olive Dieumerci, Larisa Logunova, Janette E. Bradley, Kathryn J. Else.

**Project administration:** Iris Mair, Andrew Wolfenden, Ann E. Lowe, Janette E. Bradley, Kathryn J. Else.

**Supervision:** Susanne Shultz, Janette E. Bradley, Kathryn J. Else.

**Visualization:** Iris Mair, Jonathan Fenn, Jacob Thompson.

**Writing – original draft:** Iris Mair, Jonathan Fenn, Kathryn J. Else.

**Writing – review & editing:** Iris Mair, Jonathan Fenn, Alex Bennett, Andrew Muir, Jacob Thompson, Susanne Shultz, Janette E. Bradley, Kathryn J. Else.

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
