## [Decision Letter · Decision Letter 0]

23 Dec 2023

Dear Prof Else,

Thank you very much for submitting your manuscript "The adaptive immune response to *Trichuris* in wild versus laboratory mice: An established model system in context" for consideration at PLOS Pathogens. As with all papers reviewed by the journal, your manuscript was reviewed by members of the editorial board and by several independent reviewers. In light of the reviews (below this email), we would like to invite the resubmission of a significantly-revised version that takes into account the reviewers' comments.

Academic/Associate editor comments:

I found this to be a very interesting comparison of how nematode-specific responses differ between naturally infected wild mice that have lived a life filled with a wide range of environmental exposures, co-infections, and likely multiple small intensity T. muris exposures with single challenge infected laboratory-reared mice. The manuscript is very well written and does a good job trying to tease out some of the many external and animal-host factors that could alter immune responses in wild mice. While some of the reviewers have asked for additional experimental studies to be conducted (and some of those experiments would likely be excellent), after carefully reading the manuscript and each of the reviewer’s comments, I do not think any additional experiments are required for resubmission. That stated, the study does have a number of limitations that should be addressed before publication, and this will necessitate qualifying some of the conclusions in the discussion. Please read carefully through each of the reviewer’s comments and address each of them as best as possible.

Specific critiques that I think will be especially important to address include the following:

1) Both reviewers 1 and 2 highlight that the study is a bit unbalanced in that it compares wild mice of various genetic backgrounds, ages, sexes, and exposures to lab mice that were all of one sex, one genetic background, and a single infectious exposure. This asymmetrical analysis should be highlighted as a key limitation of the study in the discussion.

2) Wild mice were dichotomized by age into young and old groups. I agree with both reviewers 1 and 2 that more detail should be provided on the parameters used to determine the age of the mice and the reliability of these approaches. Additionally, I think it would be important to know if the cutoff to determine what counts as being a young or old mouse was determined a priori.

3) I concur that the histological “gut responsiveness score” was a bit unclear to me as a reader. Would provide more details on the scoring system, whether specimens were read by a microscopist blinded to study group, and the validity of this scoring approach.

4) Finally, reviewer 1 highlights that there are numerous variables (environmental temperatures, food scarcity, reproductive state, co-infections, sex, age, frequency and intensity of T. muris exposure, etc) in the wild mice that one could attempt to replicate in experimental studies which could perhaps account for the different immune outcomes observed. I think it would be helpful for readers if the authors add a few sentences in the discussion reviewing what is known about the importance of these factors in affecting immune responses and which factors they believe are possibly key in causing wild mouse T. muris specific immune responses to be lower than responses in lab challenged mice. Additionally, it would likely also be helpful to add a line or two about approaches that possible future studies could take to better resolve the relative contribution each factor has with regards to strength and quality of immune response to T. muris.

We cannot make any decision about publication until we have seen the revised manuscript and your response to the reviewers' comments. Your revised manuscript is also likely to be sent to reviewers for further evaluation.

Sincerely,

Edward Mitre

Academic Editor

PLOS Pathogens

James Collins III

Section Editor

PLOS Pathogens

Kasturi Haldar

Editor-in-Chief

PLOS Pathogens

orcid.org/0000-0001-5065-158X

Michael Malim

Editor-in-Chief

PLOS Pathogens

orcid.org/0000-0002-7699-2064

Academic/Associate editor comments:

I found this to be a very interesting comparison of how nematode-specific responses differ between naturally infected wild mice that have lived a life filled with a wide range of environmental exposures, co-infections, and likely multiple small intensity T. muris exposures with single challenge infected laboratory-reared mice. The manuscript is very well written and does a good job trying to tease out some of the many external and animal-host factors that could alter immune responses in wild mice. While some of the reviewers have asked for additional experimental studies to be conducted (and some of those experiments would likely be excellent), after carefully reading the manuscript and each of the reviewer’s comments, I do not think any additional experiments are required for resubmission. That stated, the study does have a number of limitations that should be addressed before publication, and this will necessitate qualifying some of the conclusions in the discussion. Please read carefully through each of the reviewer’s comments and address each of them as best as possible.

Specific critiques that I think will be especially important to address include the following:

1) Both reviewers 1 and 2 highlight that the study is a bit unbalanced in that it compares wild mice of various genetic backgrounds, ages, sexes, and exposures to lab mice that were all of one sex, one genetic background, and a single infectious exposure. This asymmetrical analysis should be highlighted as a key limitation of the study in the discussion.

2) Wild mice were dichotomized by age into young and old groups. I agree with both reviewers 1 and 2 that more detail should be provided on the parameters used to determine the age of the mice and the reliability of these approaches. Additionally, I think it would be important to know if the cutoff to determine what counts as being a young or old mouse was determined a priori.

3) I concur that the histological “gut responsiveness score” was a bit unclear to me as a reader. Would provide more details on the scoring system, whether specimens were read by a microscopist blinded to study group, and the validity of this scoring approach.

4) Finally, reviewer 1 highlights that there are numerous variables (environmental temperatures, food scarcity, reproductive state, co-infections, sex, age, frequency and intensity of T. muris exposure, etc) in the wild mice that one could attempt to replicate in experimental studies which could perhaps account for the different immune outcomes observed. I think it would be helpful for readers if the authors add a few sentences in the discussion reviewing what is known about the importance of these factors in affecting immune responses and which factors they believe are possibly key in causing wild mouse T. muris specific immune responses to be lower than responses in lab challenged mice. Additionally, it would likely also be helpful to add a line or two about approaches that possible future studies could take to better resolve the relative contribution each factor has with regards to strength and quality of immune response to T. muris.

Reviewer's Responses to Questions

**Part I - Summary**

Reviewer #1: The study investigates immune responses to the helminth Trichuris muris in laboratory versus wild mice. This is an interesting and relevant topic, especially since wild mice naturally exposed to a variety of infections may have immune responses that better reflect those of adult humans – a concept that has been emerging over the last years. Less is known about how this affects responses to parasite infections. However, in the current format, this manuscript is somewhat convoluted by a number of confounding variables. While I acknowledge that working with wild mice is inherently more difficult than studying laboratory mice alone, the project could greatly improve by adding at least some more control over the multitude of factors that could be at play here.

Reviewer #2: This manuscript describes an ambitious comparative study of a laboratory model of helminth infection (i.e., inbred mice infected with the gastrointestinal nematode Trichuris muris) with a wild system in which that parasite is transmitted naturally among mice. The premise that motivates the authors – the need to understand immune responses and nematode resistance in a natural context – is compelling and important. In particular, if mouse-nematode interactions in the wild were entirely different from what is observed in controlled lab experiments, then the value, accuracy, and even the ethics of those experiments could be called into question.

By pairing a conventional experiment with field capture of naturally infected animals, and by then using the same metrics to characterize each set of mice, the authors do find some disconnects. For example, they infer that worm burdens and cytokine responses tended to be lower in the wild. Most interestingly, the authors also find that the positive association between Th1 responses and worm burden described so often in lab studies of T. muris is only observed in older wild mice. These findings suggest that although mechanisms of defense defined in lab conditions do still hold in the wild, there may be key quantitative differences and context specificities that might make wild mice better model human helminthiasis. Thus wild mice are arguably a crucial addition to the spectrum of model systems in biomedicine.

Reviewer #3: With all the possible variables at play, the application of contemporary immunological techniques to a population of wild mice, and subsequent analysis of the data, represents a complex challenge. Paradigms based on helminth infections of laboratory mice, where as many environmental and host-intrinsic variables as possible are eliminated, have dominated views of immunity to helminths for decades, even though these paradigms do not explain the complexity observed in human-helminth relationships. Some of the data presented in this manuscript do not fit neatly with the laboratory model paradigms. And yet, there are parallels in the data with trends observed in human populations. Analysis of naturally infected wild mice therefore represent a valuable opportunity to more invasively investigate helminth-host interactions to generate data that are relevant to human health.

The authors make a commendable effort to capture the relationship between host immunity and helminth infection in animals under wild/field conditions - one of the best efforts at this type of study I have seen. For the most part, appropriate methodology is employed, and the authors are appropriately circumspect in their conclusions. The manuscript is very well written and easy to read. I hope this study serves as a template for further similar investigations.

**Part II – Major Issues: Key Experiments Required for Acceptance**

Reviewer #1: While the research questions this manuscript attempts to address are interesting and relevant, it is difficult to assess the effects seen in wild mice in comparison to those in laboratory mice, as they are convoluted by multiple factors that were not excluded, including food sources and availability, stress from living in the wild (weather, predators…), continuous pregnancies of female mice. The authors acknowledge this and are making the point that this is one aspect they were trying to assess. Yet, I am somewhat concerned about how valid the conclusions from such studies can be. The genetics of wild mice are mixed. Further, the authors mention that wild mice were in many cases infected with additional helminths. The study also assesses age as a factor, by comparing juvenile to adult wild mice. But what were the parameters to determine the age of the wild mice? Are they reliable or arbitrary? (Eg. size or weight could be reduced due to food scarcity or additional environmental factors.) While the study has great potential, the conclusions would be much clearer if this multitude of factors could be controlled.

The authors may consider co-housing studies and/or controlled exposure studies, similar to many recent publications. For example, laboratory mice could be co-housed with eg. wild or pet store mice to introduce exposure to infections, whilst all other factors would still be controlled.

To exclude merely gene-/strain specific effects, multiple laboratory mouse strains could be tested against wild or co-housed mice, rather than only C57BL/6. (As long as the strains do not succumb to T.m. infection.)

It is difficult to assess whether the different response to T.m. in wild mice as compared to lab C57BL/6 is due to previous and/or constant exposure to this and/or other helminths, and/or due to microbial diversification/normalization/infection history. Experiments in Figure 2 are along those lines but could be extended. To shed further light on this, laboratory mice could be exposed to T.m. multiple times or chronically, and at distinct doses. This could be compared to T.m. infections in wild mouse co-housed laboratory mice. The wild mice could be treated for helminths prior to co-housing to exclude previous helminth exposure as a factor.

The study introduces a “gut responsiveness score”. While this is an interesting idea, it does not seem to be an established method and may need to be validated. It is unclear whether the scoring of mice was blinded or not. Since this score is an unconventional analysis method, a statistician may need to be consulted to determine whether the conclusions drawn are

---

## [Editor Report · Decision Letter 1]

13 Mar 2024

Dear Prof Else,

We are pleased to inform you that your manuscript 'The adaptive immune response to *Trichuris* in wild versus laboratory mice: An established model system in context' has been provisionally accepted for publication in PLOS Pathogens.

Best regards,

Edward Mitre

Academic Editor

PLOS Pathogens

James Collins III

Section Editor

PLOS Pathogens

Michael Malim

Editor-in-Chief

PLOS Pathogens

orcid.org/0000-0002-7699-2064
---

## [Editor Report · Acceptance letter]

8 Apr 2024

Dear Prof Else,

We are delighted to inform you that your manuscript, "The adaptive immune response to *Trichuris* in wild versus laboratory mice: An established model system in context," has been formally accepted for publication in PLOS Pathogens.

Best regards,

Michael Malim

Editor-in-Chief

PLOS Pathogens

orcid.org/0000-0002-7699-2064